# Evaluation of DNA-Launched Virus-Like Particle Vaccines in an Immune Competent Mouse Model of Chikungunya Virus Infection

**DOI:** 10.3390/vaccines9040345

**Published:** 2021-04-02

**Authors:** Jonathan O. Rayner, Jin Hyun Kim, Rosemary W. Roberts, Raphael Ryan Wood, Brian Fouty, Victor Solodushko

**Affiliations:** 1Department of Microbiology and Immunology, University of South Alabama, Mobile, AL 36688, USA; jinkim@southalabama.edu (J.H.K.); rroberts@southalabama.edu (R.W.R.); rrwood@southalabama.edu (R.R.W.); 2Departments of Internal Medicine and Pharmacology and Center for Lung Biology, University of South Alabama, Mobile, AL 36688, USA; bfouty@health.southalabama.edu; 3Department of Pharmacology and Center for Lung Biology, University of South Alabama, Mobile, AL 36688, USA; vsolodushko@southalabama.edu

**Keywords:** chikungunya virus, immune competent, mouse model, DNA vaccine, virus-like particle

## Abstract

Chikungunya virus (CHIKV) infection can result in chronic and debilitating arthralgia affecting humans in tropical and subtropical regions around the world, yet there are no licensed vaccines to prevent infection. DNA launched virus like particle (VLP) vaccines represent a potentially safer alternative to traditional live-attenuated vaccines; however, fully characterized immunocompetent mouse models which appropriately include both male and female animals for preclinical evaluation of these, and other, vaccine platforms are lacking. Utilizing virus stocks engineered to express mutations reported to enhance CHIKV virulence in mice, infection of male and female immunocompetent mice was evaluated, and the resulting model utilized to assess the efficacy of candidate DNA launched CHIKV VLP vaccines. Results demonstrate the potential utility of DNA launched VLP vaccines in comparison to a live attenuated CHIKV vaccine and identify gender differences in viral RNA loads that impact interpretation of vaccine efficacy and may have important implications for future CHIKV vaccine development.

## 1. Introduction

Chikungunya virus (CHIKV) is a mosquito-borne alphavirus (family *Togaviridae*) that is most predominantly associated with an acute febrile illness in humans. CHIKV infection is rarely fatal but in severe cases infected individuals experience severe arthralgia in the peripheral joints that can be chronic and debilitating [1,2]. The virus was first identified in Africa during an outbreak of dengue-like illness during the early 1950’s, but it is vectored by *Aedes aegypti* and *Aedes albopictus* mosquitoes and has since spread to other regions of the world where these vectors can be found including most recently the Americas as far north as Texas [3,4]. CHIKV has been associated with multiple outbreaks globally since its reemergence in Kenya in 2004 and new variants with increased fitness for the mosquito vector and atypical symptomology have emerged [5,6,7,8,9,10,11], yet the extent of CHIKV infections is underreported because the symptomology is similar to that of dengue virus (DENV) and Zika virus (ZIKV) which are transmitted by the same vectors and circulate in the same areas [12]. Thus, CHIKV poses a significant health risk throughout tropical and subtropical regions of the world, yet there are currently no licensed vaccines to prevent, or antiviral drugs to treat, infection [13].

Vaccination to prevent CHIKV infection may be necessary to protect against virus induced arthralgia and several different vaccine strategies have been evaluated including live-attenuated, subunit, virus-like particle (VLP), and DNA vaccines [13,14,15]. Live-attenuated vaccines most closely emulate a natural viral infection capable of eliciting both humoral and cellular immune responses, but they are generally considered unsafe due to the potential for reversion to virulence. Subunit vaccines and VLP vaccines which express the viral structural proteins in non-native and native forms, respectively, are considerably safer than the live-attenuated vaccines but elicit a predominantly humoral immune response that is insufficient to protect against intracellular pathogens such as CHIKV. DNA vaccines in contrast allow prolonged intracellular and membrane antigen delivery to stimulate a cellular immune response but with limited activation of the humoral immune response. Combined immunization with DNA and VLP vaccines has been demonstrated to enhance the protective immune response against other viruses [16,17]; however, an alternative strategy would be to engineer a DNA vaccine to express VLPs. A DNA launched VLP vaccine would have the potential to produce a balanced immunologic response like live-attenuated vaccines but with improved safety. Still, an important limitation of DNA vaccines is that the number of cells that are transfected following immunization is limited which in turn limits the number of antigen-producing and, more importantly, antigen presenting cells [18]. While this is less critical for developing a humoral response, this significantly limits the development of a strong cell mediated immune response.

An impediment to the development of CHIKV vaccines is the lack of fully characterized small animal models that incorporate both males and females for preclinical evaluation. Mice have been utilized to study CHIKV pathogenesis and have many attributes that are favorable for early stage testing of vaccines including the availability of large panels of reagents to quantify the host response to infection. However, CHIKV infection of immunocompetent mice, which is critical for vaccine evaluation, does not reproduce many aspects of human infections thereby limiting clinical endpoints that can be utilized for efficacy determination [19]. More recently, mutations in the CHIKV E2 glycoprotein, which represents the major viral attachment protein, and the 3′ untranslated region have been reported to enhance the virus’s virulence in immune-competent mice [20]. Utilizing a clonally derived stock of CHIKV that incorporates these mutations, infection of males and females from two commonly used immunocompetent mouse lines was evaluated and the resulting model utilized to assess the efficacy of DNA launched CHIKV VLP vaccines. Results demonstrate that two doses of DNA launched VLP vaccines can reduce viral RNA concentrations in serum and tissues following CHIKV challenge though at reduced efficiency compared to the live attenuated vaccine. It was also observed that statistically significant differences in viral RNA concentrations between vaccinated and unvaccinated animals was attributed to higher concentrations in male animals in each group, warranting continued evaluation of gender differences in response to CHIKV infection and implications on vaccine and possibly therapeutics development.

## 2. Materials and Methods

### 2.1. Cells, Viruses, Antibodies and Animals

BHK-21 cells were kindly provided by Elena Frolova (University of Alabama at Birmingham, Birmingham, AL, USA). Cells were maintained at 37 °C with 5% CO_2_ in complete media [Dulbecco’s Modified Eagle Media (DMEM) (Lonza, Basel, Switzerland) supplemented with 10% Fetal Bovine Serum (FBS) (Millipore, Burlington, MA, USA), 1% L-glutamine (Lonza) and antibiotics (100 units/mL Penicillin and 100 μg/mL Streptomycin (Lonza)]. The AF15561 and 181/25 strains of CHIKV were kindly provided by Thomas Morrison (University of Colorado, Boulder, CO, USA) and propagated on BHK-21 cells in infection media (DMEM media supplemented with 2% FBS, 1% l-glutamine and antibiotics). The morr/3′del virus was recovered by transfecting BHK-21 cells using TransIT-X2 transfection reagent (Mirus Bio, Madison, WI, USA) according to the manufacturer’s instructions. This plasmid was kindly provided by Ilya Frolov (University of Alabama at Birmingham) and is a clone of strain 181/25 engineered to generate AF15561 (nsP1 I301T, E2 I12T, E2 R82G, and 6K F42C) in addition to E2 K200R and a 3′ untranslated region (UTR) deletion (11,022–11,964) identified by Hawman et al. [20]. Master and Working stocks were confirmed to be free from mycoplasma contamination and had endotoxin levels below 0.5 endotoxin units (EU)/mL. Viral titers were determined by standard plaque assay on BHK-21 cells. Genomic titer (copies/mL) was determined by quantitative RT-PCR (qRT-PCR) using a commercially available kit (Primer Design, Southampton, UK) as described below. The morr/3′del working stock had a plaque titer of 3.5 × 10^6^ PFU/mL and a genomic titer of 3.8 × 10^7^ copies/mL (ratio of copies/PFU = 11). The AF15661 working stock had a plaque titer of 3.8 × 10^5^ PFU/mL and a genomic titer of 8.1 × 10^8^ copies/mL (ratio of copies/PFU = 2132).

Goat Anti-Mouse IgG_1_, Human ads-HRP (Cat. No: 1070-05) and Goat Anti-Mouse IgG_2a_, Human ads-HRP (Cat. No: 1080-05) antibody were from Southern Biotech (Birmingham, AL, USA). Sheep anti-mouse IgG conjugated to HRP (Cat. No NA931-1ML) was from Sigma-Aldrich (St. Louis, MO, USA). Anti-mouse CD4 conjugated to AlexaFluor 700 (Cat. No: 56-0042-80), anti-mouse CD62L conjugated to PE-Cyanine7 (Cat. No: 25-0621-82) and anti-mouse PD-1 conjugated to PE Cat. No: (Cat. No: PA5-35011) antibody were from Invitrogen (Carlsbad, CA, USA). Anti-mouse CD8a conjugated to PE-CF594 clone 53-6.7 (Cat. No: 562283), anti-mouse CD3e clone 145-2C11 conjugated to APC (Cat. No: 553066) and anti-mouse CD44 conjugated to FITC clone IM7 (Cat. No: 553133) antibody were from BD Biosciences (San Jose, CA, USA).

The C57BL/6J and BALB/c (H-2d) mice were purchased from Charles River Laboratories (Wilmington, MA, USA). For challenge studies mice were housed in polycarbonate micro-isolator cages on a ventilated rodent rack in the animal biosafety level 3 (ABSL-3) laboratory of the Laboratory of Infectious Diseases. Animal care was provided in compliance with the standard operating procedures (SOPs) of the Laboratory of Infectious Diseases, the Guide for the Care and Use of Laboratory Animals, 8th Edition [21], and the U.S. Department of Agriculture through the Animal Welfare Act (Public Law 99-198). The University of South Alabama Animal Care and Use Committee approved all mouse studies under Public Health Service assurance.

### 2.2. DNA Vaccines

Two DNA plasmid vectors encoding unmodified or modified CHIKV structural proteins needed for production of virus like particles (VLPs) were generated (Figure 1). The first DNA plasmid encoded an entire cassette of structural proteins (C-E3-E2-K6-E1) of CHIKV (strain 181/25) under control of the cytomegalovirus (CMV) promoter and terminated by a Bovine Growth Hormone (bGH) polyadenylation [poly(A)] signal (Vector A). A second DNA plasmid encoded a fusion chimeric sub-genomic CHIKV precursor in which C, E3 and the first 354 amino acids of E2 were replaced with an extracellular domain of the respiratory syncytial virus fusion (RSV-F) envelope protein (Vector B). Vector A generated a set of viral proteins C, E3, E2, 6K and E1 after post-translational cleavage of the precursor by capsid, furin and signal peptidase. Vector B generated a fusion protein comprised of the N-terminal extracellular domain (sRSV-F) of the RSV-F (RSV strain A2) protein that contains amino acids 1–529 of the un-cleaved F0 precursor of RSV-F (after furin cleavage in Golgi two amino acid chains connected by disulfide bonds were generated—a full length F2 and a truncated F1 (1–393 amino acids)) fused to a flexible linker and the C-terminal domain (truncated E2) of the CHIKV structural protein E2 (CHIKV strain 181/25; amino acids 355–423 that preserved both stem and a transmembrane domain important for its incorporation into chimeric VLPs), 6K and E1. These two vectors were used to formulate two DNA vaccines.

DNA vaccine 1 was comprised of Vector A alone and used as a control. DNA vaccine 2 was comprised by equimolar mix of Vector A and B and tested as an experimental vaccine. Both vaccines had a potential to generate VLPs due to the presence of Vector A. The experimental DNA vaccine 2 was designed to generate chimeric VLPs that in addition to E2/E1 receptors could also recognize RSV-F receptors present on the surface of other cells including dendritic, B and T cells potentially enhancing immune responses to the linked CHIKV antigens.

### 2.3. Infectious Dose Determination

For infectious dose determination, 3 groups of 6 (3 male and 3 female) C57BL/6J mice (5 to 7 weeks of age) were challenged with morr/3′del or AF15561 working stocks of CHIKV at 1 × 10^2^, 1 × 10^4^ or 1 × 10^6^ copies of genomic RNA. Each mouse received the targeted dose in a volume of 40 μl via the left hind footpad on day 0. The control group received viral diluent (phosphate buffered saline, PBS, Gibco, Thermo Fisher Scientific, Waltham, MA, USA) alone. Mice were monitored twice daily for morbidity and mortality. Body weight and body temperature were recorded daily. Body temperature was determined via an implantable transponder (BMDS, Seaford, Delaware) that was also used for animal identification. Footpad and ankle measurements were recorded daily using digital calipers (Mitutoyo, Kawasaki, Kanagawa, Japan). Blood was collected via the submandibular route on day 3 post-challenge and on day 9 post-challenge via cardiac puncture for preparation of serum and determination of viremia. On day 9 post-challenge animals were humanely euthanized and tissues (spleen, liver, ankles) were harvested for viral load determination.

To verify clinical infection in aged (14 to 16 weeks of age) BALB/c mice, a single group of 6 (3 males and 3 female) mice were challenged with CHIKV morr/3′del at a dose of 1 × 10^4^ copies/mL via the footpad as above. Animals were monitored twice daily for morbidity and mortality. Footpad and ankle measurements were recorded on days 0, 5 and 6; blood was collected on days 3 and 9; and tissues (spleen and ankles) were harvested from euthanized animals on day 9 as described above.

### 2.4. Viral Load Determination by Quantitative Reverse-Transcriptase Polymerase Chain Reaction (qRT-PCR)

Blood collected on days 3 and 9 was processed to serum using BD microtainer tubes (Fisher Scientific) according to manufacturer’s instruction. Tissues were homogenized in 2 mL tubes containing 2.4 mm metal beads (Fisher Scientific) and 0.6 mL PBS using a Bead Mill 4 (Fisher Scientific) set at 4 m/s for 40 s. The cycle was repeated 3 to 4 times and samples were iced for at least 30 s between each cycle. Particulate debris was removed by centrifugation and the supernatant was transferred to new tubes. Viral RNA was collected from serum samples using the Qiagen QiaAmp Viral RNA Mini Kit (Hilden, Germany) according to the manufacturer’s instructions. Total nucleic acid was extracted from tissue homogenates using the Zymo Quick-DNA/RNA pathogen Miniprep kit (Irvine, California) according to the manufacturer’s instructions. Viral copy number in approximately 100 ng nucleic acid extract was determined on a QuantStudio 5 real-time PCR System (Applied Biosystems, Thermo Fisher Scientific) using the GeneSig Standard CHIKV qRT-PCR assay with oasig One Step qRT-PCR master mix (PrimerDesign) according to the manufacturer’s instructions.

### 2.5. Mouse Immunogenicity and Vaccine Efficacy

For immunogenicity studies, groups of BALB/c mice were immunized with 10 μg of DNA in 25 μL PBS using a PharmaJet Injector (Golden, CO, USA), or 1 × 10^3^ plaque forming units (PFU) of live attenuated CHIKV (strain 181/25) administered subcutaneously in a 50 μl volume on two occasions separated by 3 weeks. Vaccine groups included A) Vector A alone, B) Vector A plus Vector B, and C) live attenuated CHIKV. The control group received PBS alone. Titers of total IgG, IgG_1_ and IgG_2a_ were assessed in one set of 6 mice/group starting from one month following the second dose out to 8 months via enzyme linked immunosorbent assay (ELISA). Antigen specific T cell responses were assessed in a second group of 5 mice/group two days and 5 days following stimulation with 1 × 10^4^ PFU CHIKV 181/25 (this dose was 10 times higher than what was used for immunization) two months after the last vaccination.

For efficacy studies, groups of 12 (6 male and 6 female) BALB/c mice were immunized as above then challenged with CHIKV morr/3′del on day 71 at a dose of 1 × 10^4^ copies/mL via the footpad. Animals were monitored twice daily for morbidity and mortality. Footpad and ankle measurements were recorded on days 0 and 7; blood was collected on days 3 and 9; and tissues (spleen and ankles) were harvested from animals euthanized on day 9 as described above.

### 2.6. Enzyme Linked Immunosorbent Assay (ELISA)

CHIKV specific IgG, IgG_1_ and IgG_2a_ titers were measured in serum. U-bottom 96-well ELISA plates (Nunc Maxisorp, Denmark) were coated with CHIKV strain 181/25 in PBS by direct adsorption overnight at 4 °C. Plates were washed with PBS/0.1% Tween-20 and then blocked with PBS/5% goat serum (Gibco, Carlsbad, CA, USA) for 2 h at room temperature. Serum samples were added at an initial dilution of 1:100 in triplicate, with 1:2 serial dilutions performed in PBS/1% goat serum. Plates were incubated for 1 h at room temperature, and then washed in PBS/0.1% Tween-20. A 1:10,000 dilution of HRP-conjugated anti-mouse IgG (Cat. No NA931-1ML), IgG_1_ (Cat. No: 1070-05) or IgG_2a_ (Cat. No: 1080-05) was added to the plates for 1 h at room temperature. Plates were washed and developed with 3,3′,5,5′-tetramethylbenzidine (Sigma-Aldrich, St. Louis, MO, USA) for 10 min, and stopped with 2 M HCl. The optical density of each well was measured at 450 nm on a Synergy™ 2 Multi-Mode Microplate Reader (BioTek, Winooski, VT, USA) and processed by Gen5 software.

### 2.7. Spleen Cell Subpopulations and Activation Markers

Two days and five days after stimulation, splenocytes were isolated and analyzed for cell surface activation markers. PBS-immunized mice were also stimulated and used as a control group. Immediately following euthanasia, spleens from immunized and control mice were homogenized between frosted glass slides and washed in complete medium [RPMI 1640 (Mediatech, Manassas, VI, USA) + 10% FBS + 0.2% Gentamicin (Corning, Corning, NY, USA)]. The splenocyte suspension was overlaid on Ficoll-Paque (GE Healthcare Biosciences AB, Uppsala, Sweden) and centrifuged (1000× *g*) to remove erythrocytes and dead cells. Cells recovered from the interface were washed in complete medium and counted. A total of 10^6^ spleen cells were used for each staining. To assess subpopulations and activation marker, the following anti-murine monoclonal and fluorescently tagged antibodies were used: CD4 AlexaFluor 700, CD62L PE-Cyanine7, PD-1 PE, CD8a PE-CF594, CD3e APC and CD44 FITC. Spleen cells were analyzed by BD Biosciences FACSAria cell sorter/analyzer in the University of South Alabama Flow Cytometry Core.

### 2.8. Statistical Analysis

Statistical Analysis was performed using GraphPad Prism 9 software. Data are expressed as mean ± standard deviation. Changes in body weight, body temperature, and ankle diameter in comparison to the PBS control were compared using two-way ANOVA combined with Dunnett’s multiple comparisons test. Differences in viral RNA loads between different challenge doses was analyzed via two-way ANOVA followed by Tukey’s multiple comparisons test. Differences in viral RNA loads between males and females, and between morr/3′del and AF15561 challenged animals were analyzed using two-way ANOVA combined with Śidák’s multiple comparison test. Changes in antibody titers, T cell subpopulations and activation markers were compared using two-way ANOVA combined with Tukey’s multiple comparison test. Differences in viral RNA between vaccinated and unvaccinated groups was compared using ordinary one-way ANOVA combined with Tukey’s multiple comparisons test.

## 3. Results

### 3.1. Evaluation of Clinical Endpoints for CHIKV Genetic Variants in C57BL/6J Mice

To evaluate clinical endpoints to be used for efficacy studies, a CHIKV clone (designated morr/3′del) engineered to express mutations reported to enhance virus infection in mice was compared to the parental AF15561 strain by challenging equal numbers of male and female C57BL/6J mice at doses of 1 × 10^2^, 1 × 10^4^ or 1 × 10^6^ copies of viral RNA. Over the 9-day clinical observation period, no significant changes in body weight or body temperature were observed with either CHIKV morr/3′del or AF15561 at any of the challenge doses when compared to the control group that received a sham inoculation of PBS (Figure 2). Swelling in the left ankle (proximal to the sight of challenge) and right ankle (distal to the sight of challenge) was assessed daily by measuring the width and height of each ankle and calculating the area. Animals challenged with CHIKV morr/3′del at 1 × 10^2^ copies had a significant increase in the area of the left ankle as compared to control animals on day 6 post-challenge; whereas those challenged with 1 × 10^4^ copies had significant increases on day 5, day 6 and day 8 (Figure 3a). No significant changes in the area of the left ankle were observed when the animals were challenged with the highest dose of CHIKV morr/3′del. For animals challenged with CHIKV AF15561 only the group challenged with 1 × 10^4^ copies had a significant increase in the area of the left ankle that was limited to day 7 (Figure 3b). No changes in the area of the right ankle were observed when animals were challenged with either CHIKV morr/3′del (Figure 3c) or AF15561 (Figure 3d).

Serum viral loads in infected animals were assessed after a single survival blood draw on day 3 and at euthanasia on day 9 via qRT-PCR. All animals challenged with CHIKV morr/3′del, regardless of dose, had detectable viral RNA in the serum on day 3 (Figure 4a). Peak RNA copy numbers were greatest in the 1 × 10^2^ dose group and were significantly decreased at each subsequent dose level. Only two out of six animals (1 male and 1 female) challenged with 1 × 10^2^ copies of CHIKV AF15561 had detectable RNA in serum on day 3; whereas all animals in the 1 × 10^4^ and 1 × 10^6^ dose groups had detectable levels of viral RNA. Differences in RNA copy numbers between CHIKV AF15561 dose groups were not statistically significant and, with the exception of the 1 × 10^2^ dose groups, differences in RNA copy numbers between animals challenged with CHIKV morr/3′del and AF15561 at the different dose levels were not statistically significant (Figure 4a). For both CHIKV morr/3′del and AF15561, peak RNA copy numbers were consistently lower in female animals as compared to male animals; however, the difference was statistically significant only at the 1 × 10^2^ dose level with CHIKV morr/3′del (Figure 4b) and the 1 × 10^4^ dose level with AF15561 (Figure 4c). By day 9, no viral RNA was detected in serum from animals challenged with either CHIKV morr/3′del or AF15561 (data not shown).

Viral RNA concentrations were assessed in tissues including the left ankle, spleen and liver from animals euthanized on day 9. All animals examined following challenge with CHIKV morr/3′del and AF15561 had detectable viral RNA in the left ankle (Figure 5a). Viral RNA concentrations were significantly increased in the 1 × 10^2^ dose group of CHIKV morr/3′del as compared to the 1 × 10^4^ and 1 × 10^6^ dose groups. No significant differences were observed between the 1 × 10^4^ and 1 × 10^6^ dose groups of CHIKV morr/3′del (Figure 5a). Viral RNA concentration was significantly reduced in the 1 × 10^2^ dose group of CHIKV AF15561 as compared to the 1 × 10^4^ and 1 × 10^6^ dose groups. As with CHIKV morr/3′del, there were no significant differences in viral RNA concentrations between the 1 × 10^4^ and 1 × 10^6^ dose groups of CHIKV AF15561. When comparing viral RNA concentrations in the left ankle of animals challenged with CHIKV morr/3′del versus AF15561, significantly more viral RNA was detected when comparing the 1 × 10^2^ groups while significantly less viral RNA was detected when comparing the 1 × 10^4^ dose group (Figure 5a). There were no differences in viral RNA concentration in the 1 × 10^6^ dose group of CHIKV morr/3′del versus CHIKV AF15561. Viral RNA concentrations in the left ankle of males versus females were compared and were not statistically different when animals were challenged with CHIKV morr/3′del (Figure 5b). Female animals challenged with CHIKV AF15561 had reduced concentrations of viral RNA at the 1 × 10^2^ and 1 × 10^4^ dose levels when compared to males; however, this was only significant at the 1 × 10^4^ dose level (Figure 5c).

When animals were challenged with CHIKV morr/3′del, viral RNA was detected in the spleen of all animals evaluated (Figure 6a). Viral RNA concentrations were significantly reduced in the 1 × 10^4^ dose group when compared to the 1 × 10^2^ dose group, but no significant differences were noted between the 1 × 10^2^ and 1 × 10^6^ dose groups nor the 1 × 10^4^ and 1 × 10^6^ dose groups. Viral RNA was also detected in the spleen of all animals challenged with 1 × 10^4^ and 1 × 10^6^ copies of CHIKV AF15561; however, only two males and 1 female in the 1 × 10^2^ dose group had detectable viral RNA in the spleen (Figure 6a,c). Viral RNA concentrations in the spleen of animals challenged with 1 × 10^2^ copies of CHIKV morr/3′del were significantly increased as compared to the same dose group of CHIKV AF15561 (Figure 6a); whereas no statistically significant differences were observed between the other dose groups. Decreased viral RNA concentrations were routinely observed in the spleen of female animals challenged with CHIKV morr/3′del (Figure 6b) and AF15561 (Figure 6c) when compared to males but the difference was only significant in the 1 × 10^2^ dose group of CHIKV morr/3′del. No viral RNA was detected in the liver of animals on day 9 post-challenge with either CHIKV morr/3′del or AF15561 (data not shown).

### 3.2. Verification of Clinical Endpoints in Aged BALB/c Mice

To verify clinical endpoints in aged BALB/c mice and support the DNA vaccination regimen, a single group of six mice composed equally of males and females was challenged via the footpad with 1 × 10^4^ copies of CHIKV morr/3′del while the control group received PBS alone. Based on preliminary findings in C57BL/6J mice clinical observations were limited to daily morbidity and mortality and ankle measurements on days 5 and 6. No overt clinical symptoms were noted during the 9-day observation period and there were no significant changes in the left ankle on day 5 or 6 when compared to the PBS control group (Figure 7a). Viral RNA was detected in day 3 serum samples and day 9 tissue samples (left ankle and spleen) of all BALB/c mice challenged with CHIKV morr/3′del and there were no significant differences in viral RNA concentrations in the BALB/c samples when compared to the same samples from C57BL/6J mice challenged at the same dose (Figure 7b). Viral RNA concentrations in day 3 serum samples were reduced in female BALB/c mice when compared to males; however, there were no gender differences in viral RNA load in either ankles or spleens collected on day 9 (Figure 7c).

### 3.3. Mouse Immunogenicity and Efficacy of Live Attenuated vs. DNA Vectored Vaccines

Using the PharmaJet injector, BALB/c mice were immunized with DNA vaccine 1 to generate CHIKV VLPs or DNA vaccine 2 to generate chimeric CHIKV VLPs embedded with the N-terminal extracellular domain of RSV-F. In theory, DNA vaccine 2 would permit spread of the VLP’s to other non-professional antigen presenting cells via the fusion activity of RSV-F; thereby stimulating a more robust cellular immune response. The positive control group was vaccinated with 1 × 10^3^ PFU of CHIKV 181/25 live attenuated vaccine strain. The negative control group received PBS alone. All vaccinations were delivered twice separated by three weeks. Antibody levels, including total IgG, IgG_1_ and IgG_2a_, against CHIKV were assessed monthly in a subset (6 mice per group) of vaccinated animals via ELISA starting 1 month after the second dose and continuing out to 8 months. Equal numbers of male and female animals were included in each group, but gender differences in the immune response were not assessed due to group size limitations. Two doses of the live attenuated CHIKV vaccine resulted in the highest levels of total IgG, IgG_1_ and IgG_2a_ antibody production but titers declined significantly by the 5th month (Figure 8). In contrast, two doses of the DNA vaccines resulted in modest antibody levels that persisted out to 6 months (Figure 8a). IgG_1_ antibody titers were elevated as soon as 1 month following the second dose of DNA vaccination and remained consistent over a 6-month period (Figure 8b); whereas IgG_2a_ antibody titers took two months to rise significantly above baseline and fluctuated at subsequent samplings (Figure 8c). No significant differences in antibody titers were observed with the two DNA vaccines.

A second group of vaccinated mice (5 mice/group) were stimulated with 1 × 10^4^ PFU of CHIKV 181/25 and splenocytes were isolated 2- and 5-days post-stimulation to assess cell surface activation markers. PBS vaccinated mice were also stimulated and used as controls. Figure 9 shows the results when differences were identified. Day 0 represents data from unstimulated mice. CHIKV stimulation increased the percentage of CD4+ cells and decreased the percentage of CD8+ cells in all animals, but especially in mice vaccinated with live attenuated CHIKV 181/25 vaccine (Figure 9a,b). This suggests that CHIKV infection induces a Th2 immune response in this model. Neither DNA vaccine was different from the non-immunized control. Only mice immunized with the live attenuated vaccine increased the percentage of activated (CD44+/CD62l-) CD4+ cells 5 days after stimulation (Figure 9c); whereas the percentage of activated CD8+ cells following stimulation decreased in all groups but most dramatically in the groups that had received the DNA vaccines (Figure 9d). Lastly, the percentage of PD-1 positive cells following stimulation within the population of activated CD4+ cells (Figure 9e) and CD8+ cells (Figure 9f) were increased in the DNA vaccinated animals when compared to the animals that received the live attenuated vaccine and the unvaccinated controls. The increase was most significant in the group that received DNA vaccine 2.

The third group of vaccinated animals (12 mice/group) were challenged with 1 × 10^4^ copies of CHIKV morr/3′del 71 days following the first vaccination and clinical progression was assessed over a 9-day observation period. No significant clinical findings were observed in any of the animals, including the left ankle, consistent with preliminary studies in aged BALB/c mice. All animals in the unvaccinated control group had detectable viral RNA in the serum on day 3 post-challenge as compared to 17% in the group vaccinated with CHIKV 181/25, 83% in the group vaccinated with DNA vaccine 1, and 50% in the group vaccinated with DNA vaccine 2 (Figure 10a). Consequently, viral RNA concentration in serum on day 3 was significantly reduced in animals vaccinated with CHIKV 181/25 (*p* = 0.0021) and moderately reduced in the group that received DNA Vaccine 2 (*p* = 0.022) when compared to viral RNA titers in the unvaccinated animals using ordinary one-way ANOVA. Viral RNA copy number in serum was unaffected in the group that received DNA vaccine 1 when data from males and females was considered together. When considering day 3 serum viral RNA concentrations from male and female animals separately using two-way ANOVA the male animals in each vaccine group, including DNA vaccine 1, had significantly reduced titers (*p* ≤ 0.0005) when compared to unvaccinated animals (Figure 10b). Viral RNA copy number in day 3 serum from female animals was unaffected by vaccination because the copy number was already significantly reduced in female animals as compared to male animals in the unvaccinated control group (*p* = 0.0008).

On day 9 post infection, 100% of unvaccinated mice had detectable quantities of viral RNA in the left ankle and spleen (Figure 11a,c, respectively). Viral RNA concentrations were similar in the left ankle of unvaccinated animals when males and females were compared but were significantly reduced in the spleens of female animals as compared to the male animals (Figure 11b,d, respectively). Viral RNA was also detected in the left ankle of 100% of mice vaccinated with DNA vaccine 1, 92% of mice vaccinated with DNA vaccine 2, and 42% of mice vaccinated with CHIKV 181/25 (Figure 11a). Viral RNA concentrations in the left ankle were significantly reduced in all three vaccine groups when compared to the unvaccinated control animals; however, this reduction was attributed predominantly to the male animals in each group (Figure 11b). Viral RNA was detected in 83% and 92% of spleens collected on day 9 from mice vaccinated with DNA vaccine 1 and DNA vaccine 2, respectively, but only 8% of animals vaccinated with CHIKV 181/25 (Figure 11c). Viral RNA concentrations in the spleens were significantly reduced in all three vaccine groups as compared to the unvaccinated animals but most significantly in the group vaccinated with CHIKV 181/25 and DNA vaccine 2. The reduction in viral RNA concentration was attributed predominantly to the male animals in each group as there was no difference in viral RNA concentration when comparing female animals in the vaccinated and unvaccinated groups.

## 4. Discussion

Human infections with CHIKV are of increasing concerns in tropical and subtropical regions of the world and have the potential to significantly impact quality of life. New diagnostic test have been developed to differentiate CHIKV infections from other common mosquito borne viruses such as DENV and ZIKV that circulate in these regions [22,23]; and can be utilized to target vaccination campaigns to areas that need it most. To this end, next generation vaccines which are both safe and effective are desperately needed to prevent infection and chronic disease associated with CHIKV worldwide. DNA launched VLP vaccines which offer the potential to elicit a balanced immunological response required for obligate intracellular pathogens such as CHIKV represent a safer alternative to traditional live-attenuated vaccines. Essential to the preclinical evaluation of these vaccines is the availability of immune-competent animal models which accurately recapitulate the human disease. For a myriad of reasons, which includes the availability of reagents to track the response to infection, mice are often used as a first-tier model for vaccine evaluation; however, in the case of CHIKV, infection of immune competent mice such as C57BL/6J and BALB/c is limited to a transient viremia and biphasic swelling in the ankle proximal to the sight of challenge thereby limiting clinical endpoints that can be used for determining vaccine efficacy [19]. More recently, mutations in the E2 glycoprotein and 3′ UTR of CHIKV have been reported to enhance virulence in immune-competent mice; however, these studies were limited to juvenile mice of 3 to 4 weeks of age which would not support most vaccination regimens [20]. Additionally, the National Institutes of Health in the United States expects that sex be included as a biological variable when designing and reporting results from vertebrate animal studies [24], yet gender differences in response to CHIKV infection or vaccination in mice have not been reported elsewhere.

To address these issues and establish a challenge dose to be used for evaluation of DNA launched VLP vaccine candidates, a working stock of CHIKV established from an infectious clone engineered to express virulence determinants in the E2 Glycoprotein and 3′ UTR, designated morr/3′del, was utilized to challenge groups of mice at increasing concentrations. Adult, 5 to 7 weeks of age, C57BL/6J mice were examined first and each group consisted equally of males and females to address gender differences. In contrast to previous reports utilizing juvenile C57BL/6J mice, no significant clinical observations, including changes in body weight and viral dissemination indicated by swelling in the right ankle, were noted over the 9-day clinical observation period. Transient swelling in the left ankle between days 5 and 8 was observed but is not uncommon for CHIKV infection of immune-competent mice with or without these mutations [19]. Interestingly, swelling was only significant in the low (1 × 10^2^) and median (1 × 10^4^) dose groups and most frequently observed with the median dose.

Due to restrictions on the volume of blood that can be humanely drawn from mice, clinical sampling during the observation period was limited to a single survival blood collection on day 3 post challenge in addition to blood and tissues on day 9 following euthanasia. Viral RNA was detected in the serum of all animals challenged at all three doses on day 3; however, viral RNA concentrations were significantly higher in the low (1 × 10^2^) dose group as compared to either the median (1 × 10^4^) or high (1 × 10^6^) dose groups. By day 9, the virus had cleared from the blood, but viral RNA remained detectable in both the left ankle and the spleen. Viral RNA concentrations again remained significantly elevated in the lowest dose group as compared to the median and high dose groups. Together these results suggest that the virus is cleared from the blood and tissues more rapidly at higher doses and is likely due to the immunological response. Additionally, it is unclear why swelling in the left ankle was absent from animals that received the highest challenge dose given that viral RNA remained detectable in the left ankle at comparable concentrations to the median dose group. The immunological response also likely plays a role in this observation and a natural history study that incorporates serial sacrifice to allow more clinical sampling will be required to fully address virological, immunological, and pathological responses in mice at each of the challenge doses.

Given the absence of clinical symptoms previously associated with mutations in the E2 glycoprotein and 3′ UTR of CHIKV on mice, the experiment was repeated using the parental AF15561 strain of CHIKV to independently assess if these mutations are associated with increased virulence in mature immunocompetent mice and validate the use of the mutant virus for further preclinical model development. Equivalent copy numbers were used for this comparison because peak infectious titers were reduced by one log with CHIKV AF15561 as compared to CHIKV morr/3′del, while viral RNA concentrations remained elevated giving an approximately 200-fold difference in particle to PFU ratios between the two stocks. As expected, no significant changes in clinical symptoms were observed during the 9-day observation period and, in contrast to animals challenged with CHIKV morr/3′del, swelling in the left ankle was limited to day 7 post challenge. Additionally, viral RNA was detected less frequently and at decreased levels in both day 3 serum and day 9 tissues when comparing the 1 × 10^2^ dose group of CHIKV AF15561 and CHIKV morr/3′del. In consideration of these differences alone, it is reasonable to conclude that mutations in the E2 glycoprotein and 3′ UTR of CHIKV do confer increased virulence and that differences in clinical observations between this and previous studies are likely related to the age of the animal. Alternatively, it can be argued that the difference in response at the low dose is related to the significantly decreased infectious dose of CHIKV AF15561 as compared to CHIKV morr/3′del. These questions would also require more comprehensive natural history studies to be fully addressed.

BALB/c mice are another commonly used immune-competent mouse strain and were employed to further validate the optimized model conditions which included a challenge dose of 1 × 10^4^ copies of CHIKV morr/3′del along with day 3 serum and day 9 tissue viral RNA loads. In this case, mice were 14 to 16 weeks of age to correspond with the DNA launched VLP vaccine regimen, and ankle measurements were limited to days 5 and 6, when swelling was expected, to minimize stress on the animals and risk to the laboratorians handling the animals. Viral RNA was detected in serum on day 3 and tissues on day 9 of all BALB/c mice at similar levels to that seen with the same challenge dose in C57BL/6J mice; however, swelling of the left ankle on day 5 and 6 were not observed. Ankle measurements are subject to a great deal of variability from day to day and user to user as can be seen from the error bars presented in Figure 3 and Figure 7; thus, ankle swelling is not the most reliable clinical endpoint. However, it also cannot be ruled out from these experiments that age of the mice and/or mouse strain plays a role in this observation. Additionally, it was consistently observed with both CHIKV morr/3′del and AF15561 in both C57BL/6J and BALB/c mice that viral RNA concentrations were reduced in female animals as compared to male animals. The observation was most consistently observed in day 3 serum samples though the difference was not always statistically significant.

Upon verification of the challenge dose and clinical endpoints, BALB/c mice were immunized twice with DNA launched VLP vaccines versus a live attenuated CHIKV vaccine strain and the immunological response was tracked prior to challenge. Following the two-dose regimen, B-cell and T-cell responses were elicited by both DNA launched VLP candidates though antibody titers and activated antigen-specific CD4 cells were significantly reduced in comparison to the live attenuated vaccine. Consequently, the live attenuated vaccine was most effective at reducing or eliminating viral RNA concentrations in serum on day 3 and tissues on day 9; however, vaccination with the DNA launched VLP vaccines did significantly reduce viral RNA concentrations. Once again viral RNA concentrations in the serum of unvaccinated mice were significantly reduced in females as compared to males and with the increased group sizes significant decreases were also observed in the spleen of female animals on day 9 but not the ankles. Furthermore, when comparing RNA concentrations between the unvaccinated and vaccinated groups significant reductions were attributed solely to female animals. Several epidemiological studies have indicated gender disparities in response to CHIKV infection that are likely attributable to differential immunological responses [25,26,27]. Gender differences in the immunological response were not evaluated given group size limitations, but the consistent differences in gender response here combined with epidemiological reports support continued assessment of gender and age differences following CHIKV infection. These differences will likely have important implications for continued vaccine and therapeutics development.

Vaccination with DNA vaccine 2, which incorporated a fusion of the E2 protein with the ectodomain of RSV-F protein was most effective at reducing viral RNA concentrations in clinical samples. RSV-F is a transmembrane envelope protein which can, by itself, mediate virus-to-cell, and cell-to-cell fusion. The ectodomain of RSV-F has been shown to be sufficient for it to attach to cell membranes and enter target cells [28], thus the objective with DNA vaccine 2 was to generate a chimeric VLP better capable of dispersing from the originally transfected cell to multiple neighboring cells thereby increasing the number of cells presenting antigen in association with MHC-I and boosting the T-cell response compared to traditional DNA vaccines [29,30]. Except for activation of PD-1 positive cells, significant differences in the immune response measured here to DNA vaccine 1 versus DNA vaccine 2 were minimal. Additional immunological endpoints, prior to and following challenge, would have to be considered to explain this observation.

Finally, these experiments were intended to provide preliminary proof of concept for the DNA launched VLP vaccine strategy and while the attenuated vaccine remained superior, different dosing regimens and DNA constructs that incorporate the RSV-F fusion technology to safely boost the immunological response and protective efficacy of these vaccines should be considered for continued development.

## Figures and Tables

**Figure 1 vaccines-09-00345-f001:**
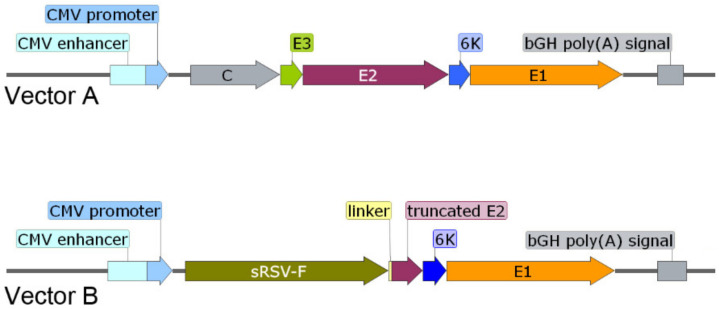
Chikungunya virus DNA vaccine vector constructs. Vector A expresses the CHIKV structural protein open reading frame under the control of a CMV promoter and followed by the bGH poly(A) signal. Vector B encodes a fusion between sRSV-F and the CHIKV E2 glycoprotein and the remaining glycoprotein region under the control of a CMV promoter and followed by the bGH poly(A) signal.

**Figure 2 vaccines-09-00345-f002:**
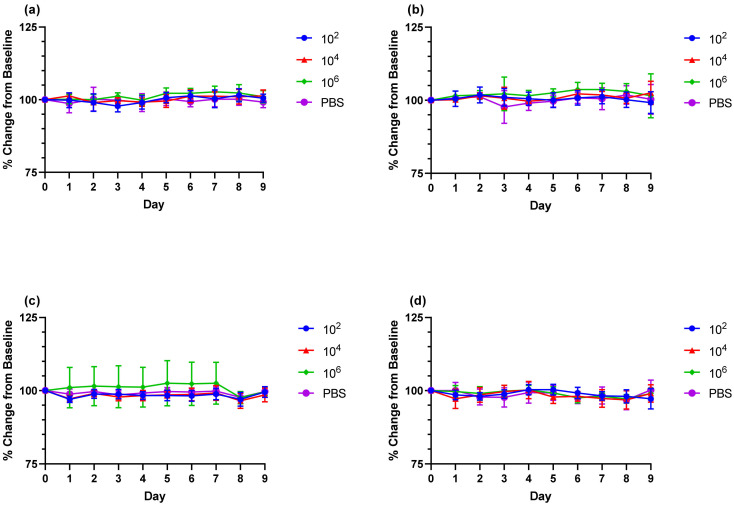
Changes in body weight and body temperature in C57BL/6J mice challenged with CHIKV morr/3′del vs. AF15561 viruses. Percent change in body weight from baseline (day 0) following challenge with CHIKV morr/3′del (**a**) or CHIKV AF15561 (**b**). Percent change in body temperature from baseline following challenge with CHIKV morr/3′del (**c**) or CHIKV AF15561 (**d**).

**Figure 3 vaccines-09-00345-f003:**
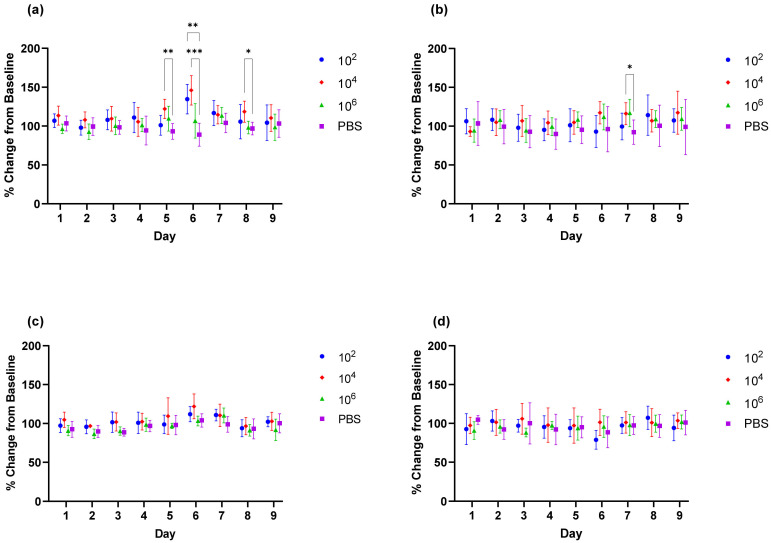
Changes in the left ankle and right ankle of C57BL/6J mice challenged with CHIKV morr/3′del vs. AF15561. Percent change in the left ankle from baseline (day 0) following challenge with CHIKV morr/3′del (**a**) and CHIKV AF15561 (**b**). Percent change in the right ankle from baseline following challenge with CHIKV morr/3′del (**c**) and CHIKV AF15561 (**d**). * *p* < 0.05, ** *p* < 0.005, *** *p* < 0.0008.

**Figure 4 vaccines-09-00345-f004:**
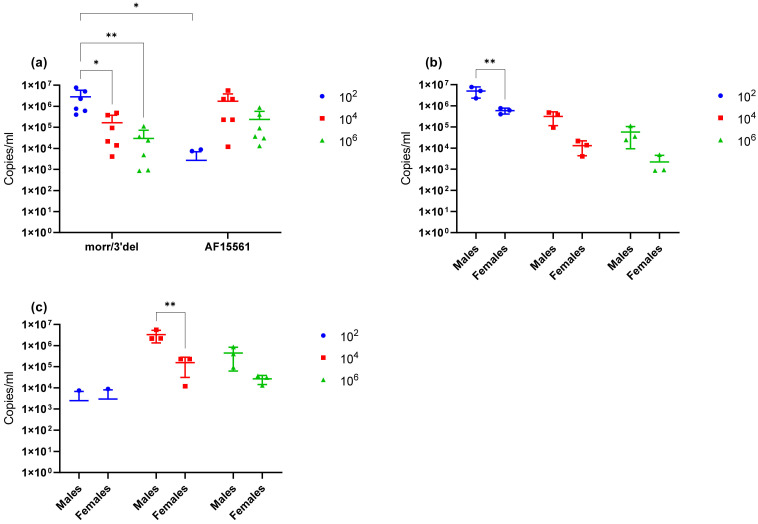
Serum viral loads in animals challenged with CHIKV morr/3′del and AF15561. Values are expressed as copies/mL of serum. (**a**) Viral RNA copy number in serum on day 3 post-infection with CHIKV morr/3′del and AF15561 at 1 × 10^2^, 1 × 10^4^ and 1 × 10^6^ dose levels. (**b**) Viral RNA copy numbers in male versus female animals challenged with CHIKV morr/3′del. (**c**) Viral RNA copy numbers in male versus female animals challenged with CHIKV AF15561. * *p* < 0.05, ** *p* < 0.005.

**Figure 5 vaccines-09-00345-f005:**
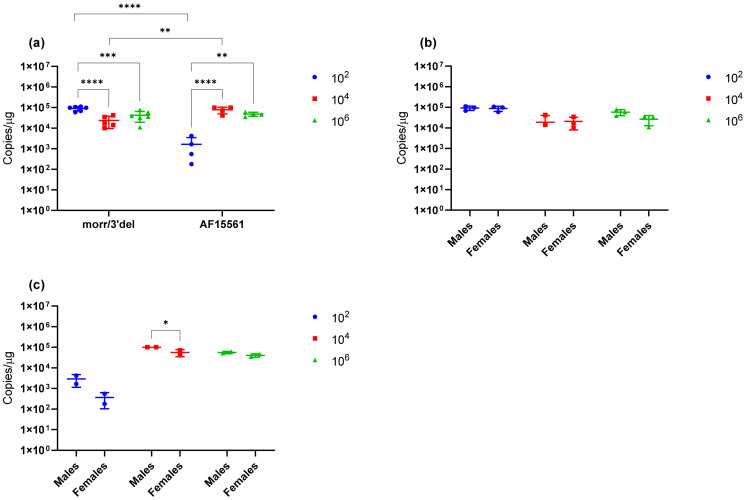
Viral RNA concentrations in the left ankle of animals challenged with CHIKV morr/3′del and AF15561. Values are expressed as copies/μg RNA. (**a**) Viral RNA copy number in the left ankle on day 9 post-infection with CHIKV morr/3′del and AF15561 at 1 × 10^2^, 1 × 10^4^ and 1 × 10^6^ dose levels. (**b**) Viral RNA copy numbers in male versus female animals challenged with CHIKV morr/3′del. (**c**) Viral RNA copy numbers in male versus female animals challenged with CHIKV AF15561. * *p* < 0.05, ** *p* < 0.005, *** *p* < 0.0008, **** *p* < 0.0001.

**Figure 6 vaccines-09-00345-f006:**
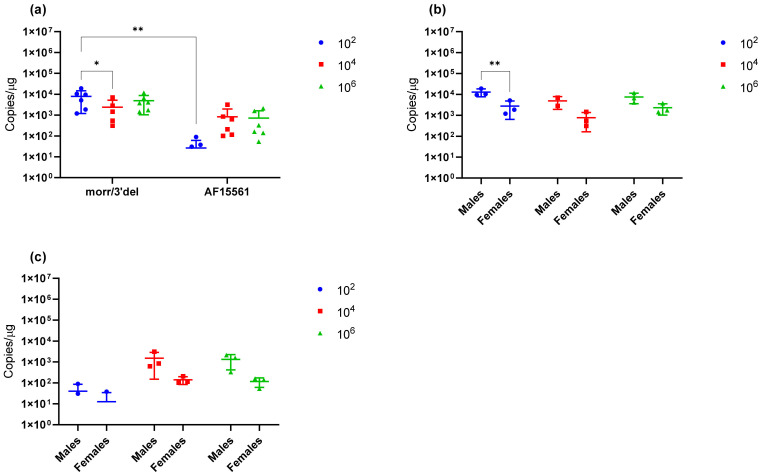
Viral RNA concentrations in the spleen of animals challenged with CHIKV morr/3′del and AF15561. Values are expressed as copies/μg RNA. (**a**) Viral RNA copy number in the spleen on day 9 post-infection with CHIKV morr/3′del and AF15561 at 1 × 10^2^, 1 × 10^4^ and 1× 10^6^ dose levels. (**b**) Viral RNA copy numbers in male versus female animals challenged with CHIKV morr/3′del. (**c**) Viral RNA copy numbers in male versus female animals challenged with CHIKV AF15561. * *p* < 0.05, ** *p* < 0.005.

**Figure 7 vaccines-09-00345-f007:**
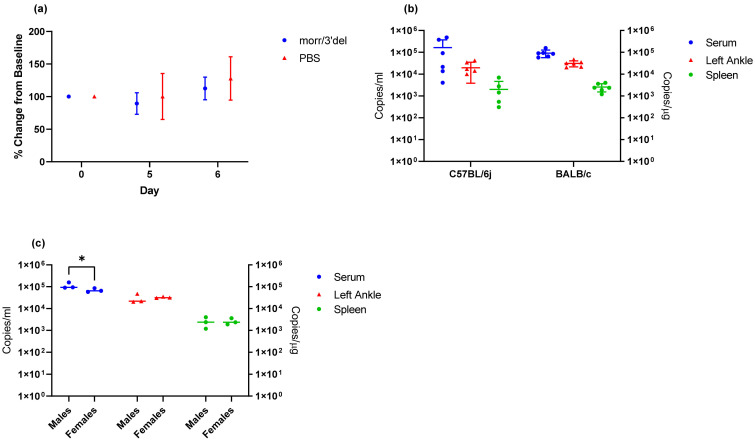
Clinical endpoints in aged BALB/c mice challenged with CHIKV morr/3′del at 1 × 10^4^ copies. (**a**) Percent change in the left ankle on days 5 and 6 following challenge. (**b**) Viral RNA copy number in serum collected on day 3, left ankle collected on day 9, and spleen collected on day 9 in C57BL/6J mice versus BALB/c mice. (**c**) Viral RNA copy numbers in serum and tissues of male versus female BALB/c mice. Values are expressed as copies/mL for serum and copies/μg RNA for tissues. * *p* < 0.05.

**Figure 8 vaccines-09-00345-f008:**
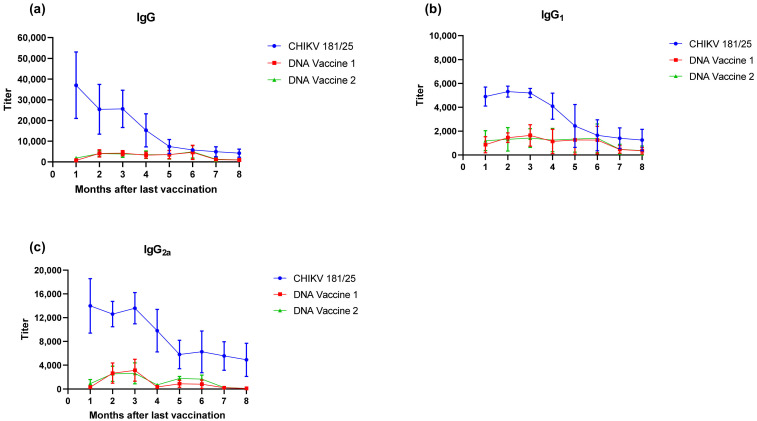
CHIKV antibody titers in BALB/c mice following two doses of DNA vaccine 1, DNA vaccine 2 or the CHIKV 181/25 live attenuated vaccine. (**a**) Total IgG antibody titers over the 8-month study period. (**b**) IgG_1_ antibody titers over the 8-month study period. (**c**) IgG_2a_ antibody titers over the 8-month study period.

**Figure 9 vaccines-09-00345-f009:**
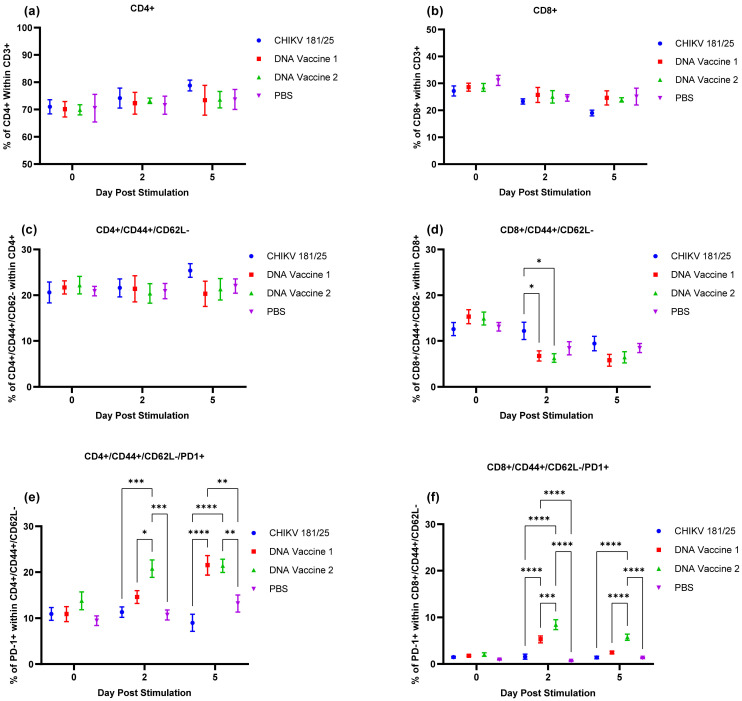
Percent of antigen-specific T cells at baseline and 2- or 5-days post-stimulation with CHIKV 181/25. (**a**) Percent of CD4+ T cells within the total (CD3+) T cell population. (**b**) Percent of CD8+ T cells with the total T cell population. (**c**) Percent of activated (CD44+/CD62l-) cells within the population of CD4+ cells. (**d**) Percent of activated (CD44+/CD62l-) cells within the population of CD8+ cells. (**e**) Percent of PD-1 positive cells within the population of activated CD4+ cells. (**f**) Percent of PD-1 positive cells within the population of activated CD8+ cells. Control = mice immunized with PBS; DNA 1 = mice immunized with DNA vaccine 1; DNA 2 = mice immunized with DNA vaccine 2; CHIKV = mice immunized with the live attenuated CHIKV 181/25 strain. * *p* < 0.05, ** *p* < 0.005, *** *p* < 0.0008, **** *p* < 0.0001.

**Figure 10 vaccines-09-00345-f010:**
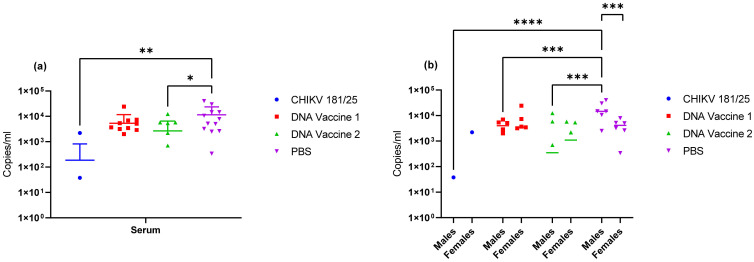
Viral RNA concentrations in serum collected on day 3 post-challenge. (**a**) Viral RNA copy number in vaccinated versus unvaccinated animals. (**b**) Viral RNA copy number in male versus female animals. * *p* < 0.05, ** *p* < 0.005, *** *p* < 0.0005, **** *p* < 0.0001.

**Figure 11 vaccines-09-00345-f011:**
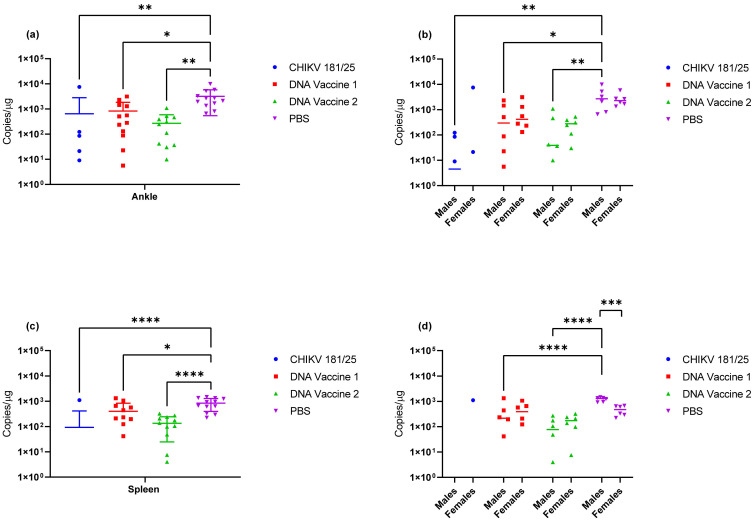
Viral RNA concentration in tissues collected on day 9 post-challenge. (**a**) Viral RNA copy number in the left ankle of vaccinated versus unvaccinated animals. (**b**) Viral RNA copy number in the left ankle of male versus female animals. (**c**) Viral RNA copy number in the spleen. (**d**) Viral RNA copy number in the spleen of male versus female animals * *p* < 0.05, ** *p* < 0.005, *** *p* < 0.0008, **** *p* < 0.0001.

## Data Availability

Data is contained within the article.

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
