# Peer review of "Evaluation of DNA-Launched Virus-Like Particle Vaccines in an Immune Competent Mouse Model of Chikungunya Virus Infection"

_vaccines, 2021, doi:10.3390/vaccines9040345_

Round 1

Reviewer 1 Report

The authors attempted to evaluation DNA-launched virus-like particle vaccines in an immune competent mouse model of Chikungunya virus infection. The topic is important and the manuscript is well-written.

In the title: Please change "DNA launched" into "DNA-launched" and "virus like" into "virus-like".

In the abstract, please explicitly indicate the gender-specific results that you are alluding to. 

In the results section, please clarify the significance of these p values in Fig 10 . * p<0.05, **p<.005, ***p<.0008, ****p<.0001

In the discussion, please reference some diagnostic tests for Chikungunya virus infection (PMID: 33167379 and PMID: 28606643). Explain how diagnosis and vaccination efforts can be coordinated to ensure that vaccination campaigns can be better targeted to needy areas. 

Reviewer 2 Report

The authors have written the paper well in terms of vaccine development aspect but it is suggested to highlight or elaborate more global importance and new trend of clinical manifestation of chikungunya fever.

Chikungunya is a re-emerging virus that has a risk to spread globally. Chikungunya disease does not often result in death, but the symptoms can be severe and disabling for months. Another challenges is that the symptoms of chikungunya are similar to those of dengue and Zika, diseases spread by the same mosquitoes that transmit chikungunya. The reviewer has noticed active role of Aedes albopictus in CHIKV transmission in rural population during Maldives outbreak in 2008 which is epidemiologically important as they have adaptability to changing climatic conditions.  

Old  strains of Chikungunya virus may cause clinical disease for a week. During the La Reunion outbreak in 2006, up to 60% of patients reported prolonged painful joints three years following initial infection.

Schilte C, Staikowsky F, Staikovsky F, Couderc T, Madec Y, Carpentier F, et al. (2013). "Chikungunya virus-associated long-term arthralgia: a 36-month prospective longitudinal study". PLOS Neglected Tropical Diseases. 7 (3): e2137. doi:10.1371/journal.pntd.0002137. PMC 3605278. PMID 23556021.

The authors have critically analysed the limitation of attenuated vaccine as it is reactive and there is a potential chance of reversion of virulence. There is no doubt that we need to move forward with alternative vaccine development strategies. The R&D for Covid-19 has accelerated fast track application of new technology to deal with global pandemic and this paper may help readers and policy makers to think seriously. New vaccine technology such  use of virus-like particles (VLPs) instead of attenuated virus was already in pipeline. The authors have elaborated the impediments to the development of CHIKV vaccine as well as limitation of the study design and pre-clinical evaluation which is appreciable. It will be good to shorten description of materials and methods. Concise presentation of results would be appreciated. The last paragraph under discussion may be divided into two, i.e. concluding remarks - Finally, these experiments were intended...    

Reviewer 3 Report

The development of a vaccine against chikungunya would be a decisive asset in the fight against this disease. Using a mouse model, the authors have shown the possibilities and limits of using a DNA vaccine. They have taken into account the current issues, in particular the fact of comparing the immune response according to the sex of the subjects.
This work is remarkable with a detailed description of the methodology and a clear presentation of the results. The discussion is concise and highlights the aspects to be improved
